

# Wildfire effects on ecosystem nitrogen cycling in a Chinese boreal larch forest, revealed by [15]N natural abundance

Weili Liu[1,2], Lin Qi[1], Yunting Fang[1], Jian Yang[1,3]*

[1]State Key Laboratory of Forest and Soil Ecology, Institute of Applied Ecology, Chinese Academy of

Sciences, Shenyang 110164, P. R. China

[2]University of Chinese Academy of Sciences, Beijing 100049, P. R. China

[3]Department of Forestry, University of Kentucky, Lexington, KY, 40546, USA

*Correspondence to: J. Yang (jian.yang@uky.edu)

**Abstract.** Wildfire is reported to exert strong influences on N cycling, particularly during the early

succession period immediately after burning (i.e., < 1 year). Previous studies have mainly focused on

wildfires influences on inorganic N concentrations and N mineralization rates; but plant and soil [15]N

natural abundance (expressed by $\delta^{15}N$), as a spatial-temporal integrator of ecosystem N cycling, could

provide a more comprehensive understanding of wildfire on various N cycling processes at a relatively

broader time scale. In this study, we attempted to evaluate legacy effects of wildfire on nitrogen cycling

using $\delta^{15}N$ in a boreal forest of northeastern China, which is an important yet understudied ecosystem.

We measured inorganic N concentrations ($NH_4^+$ and $NO_3^-$) and net N transformation rates (net

ammonification, net nitrification, and net mineralization) of organic and mineral soil 4 years after a

wildfire and compared with unburned area. We also measured $\delta^{15}N$ of plant and soil samples in 4 and 5 years after the fire. We found that even 4 years after burning, wildfire still increased net mineralization and net ammonification in the organic soil and increased $NH_4^+$ and total inorganic N (TIN) concentrations in the mineral soil. Organic soil and foliar $\delta^{15}N$ were significantly higher (by 2.2‰ and 7.4‰, respectively) in the burned area than the unburned area. Five years after fire, plant tissues such as foliar, branch, fine roots and moss in the burned area were increased significantly (by 1.7‰ to 6.4‰) greater than that in unburned area. The wildfire also significantly increased the $\delta^{15}N$ of Oi, Oa+e and 0-10 cm mineral soil, but had no significant effects on deeper layer of mineral soil. These results indicate the wildfire had a strong legacy effect on N cycling. We suggest that the change of abiotic environment was the primary mechanism determining inorganic nitrogen transformation rates, and the $NH_3$ volatilization might play a key role in severe N losses and thereby affect soil and plant $^{15}N$ in this ecosystem.

## 1 Introduction

Wildfire-induced nitrogen (N) cycling changes can greatly alter ecosystem structure and functions, such as species composition and biodiversity (Gallant *et al*., 2003), biogeochemical cycles and productivity (Boerner, 1982; Chorover *et al*., 1994; Woodmansee and Wallach, 1981), as N is likely to be the most essential element limiting plant growth in terrestrial ecosystems (Popova *et al*., 2013; Stark and Hart,

1997). In response to recent global climate changes, wildfire frequency and extent in temperate and

boreal forests are projected to be enhanced (Flannigan *et al*., 2009; Lucash *et al*., 2014; Westerling *et al*.,

2006). Therefore, a better understanding of wildfire effects on N dynamics is of growing importance.

Many studies have attempted to examine the effects of fire on N cycles through the analysis of available

N concentrations and N mineralization rates (e.g., Turner *et al*., 2007; Koyama *et al*., 2010; Deluca *et al*., 2006). However, two principal limitations have greatly challenged this objective. First, these two

indices vary significantly in space and time (Cain *et al*., 1999; Hu *et al*., 2013). Second, soil available N

concentrations and N mineralization rates only account for a fraction of N cycling processes. Other

N-related processes such as denitrification and leaching are also important in forest ecosystems (Fang *et al*., 2015), yet these processes are difficult to measure directly, thus constraining our ability to

generalize the response of N cycle to wildfire.

The natural abundance of $^{15}N/^{14}N$ of plant and soil is considered as a more time- and space- integrator

of N cycling than available N concentration and N mineralization rate and could reflect the openness of

ecosystem N cycling (Robinson, 2001). Soil processes, such as N mineralization, nitrification,

denitrification and $NH_3$ volatilization, discriminate against $^{15}N$ and lead to soil N pool with different

$\delta^{15}N$ signatures (Craine *et al*., 20015; Houlton *et al*., 2007; Vallano and Sparks, 2013), which further

express in the values of plant $\delta^{15}N$ that utilize these N pools for their N demands. Higher values of $^{15}N$

in soil and plant generally indicate larger N losses through ammonia volatilization, nitrification or

denitrification (Craine *et al.*, 2009; Houle *et al.*, 2014; Matsushima *et al.*, 2012). Thus, the ecosystems

with a more open N cycle tend to be isotopically enriched in [15]N. In fact, although the responses of

available N concentration and N mineralization to wildfire could vary by time and space, we can expect

such ecosystems would always become more N open when available N supply exceeds demand,

resulting in an increase in N loss and higher values of $\delta^{15}N$ in plant and soil. Therefore, $\delta^{15}N$ could

provide us a promising and comprehensive tool to detect the effect of wildfire on N cycling.

Boreal forests are typical N limited ecosystems where N cycling is slow and a large proportion of

total N capital is tied up in undecomposed organic matter (Hyodo *et al.*, 2013; Metcalfe *et al.*, 2013;

Popova *et al.*, 2013; Sah *et al.*, 2006). This is consistent with low $\delta^{15}N$ values of soil and plant (Hogberg,

1997). Wildfire is a primary agent of disturbance in boreal forests and has a profound impact on N

cycling (Baird *et al.*, 1999; Bond-Lamberty *et al.*, 2006). Wildfires consume N from vegetation and the

upper surface soil layer, resulting in a reduction of N storage in burned forest (Hogberg, 1997; Hyodo *et

al.*, 2013). Estimates of post-fire inorganic N concentrations and mineralization rates vary, but most

studies show an immediate increase in inorganic nitrogen (Deluca and Sala, 2006; Koyama *et al.*, 2012;

Turner *et al.*, 2007). Nevertheless, the immediately increased $NH_4^+$ can decline to the pre-fire level

within the first year and the elevated $NO_3^-$ generally returned to pre-fire level within 5 years (Wan *et al.*

2001). Studies have also shown a wildfire-induced pulse, characterized by a sharp increase in foliar

$\delta^{15}N$ (LeDuc *et al*., 2013). However, this wildfire-induced pulse in foliar $\delta^{15}N$ is shown short-lived

followed by a rapid decline after the first few years (LeDuc *et al*., 2013). Compared to much attention

paid to these changes in N dynamics at the 0-3 years after a fire event in boreal forest (Baird *et al*., 1999;

DeLuca and Zouhar, 2000; White, 1986), little is known about the legacy effects of fires on N cycling

following this pulse, particularly through the use of natural abundance of N isotopes to detect the

responsible processes.

The Great Xing'an Mountains of northeast China are located on the southern extension of the larch

forests of the eastern Siberia. It has been reported that the boreal forests in Northeast China stored 1.0 -

1.5 Pg C and provided approximately 24 - 31% of the total timber production in China (Fang *et al*.,

2001). This region experiences frequent wildfires with historical fire return intervals of 30-120 years.

Despite the knowledge that fires can significantly influence soil N dynamics elsewhere, the

understanding on the influence of post-fire N cycling in this region is limited. A better understanding of

wildfire effects on these boreal larch forests could fill this gap. Previous studies have investigated the

responses of soil inorganic N concentration and N mineralization to wildfire in this region (Kong *et al.,*

2015), and showed that a pulse of inorganic nitrogen concentration appeared in one year post-fire.

However, there are few studies of response of soil inorganic N and mineralization to fire in this boreal

larch forest over a relatively longer post-fire time period, and even less regarding the fire effects on plant and soil $\delta^{15}$N.

In this study, we compared the N status of burned area (4 and 5 years after a large wildfire) with unburned area through comprehensive analysis of $\delta^{15}$N for foliage and soil, inorganic nitrogen concentrations and N mineralization rates. Our overarching goal was to determine the legacy effect of the wildfire on N cycling in boreal forest of Greater Xing'an Mountains and explore the underlying mechanism. Specifically, we expected that:

1) The inorganic nitrogen concentration and N mineralization rate in the burned sites would be similar with the unburned area since most studies have shown that available N concentrations and mineralization rates declined to the pre-fire level within 5 years after fire;

2) Soil $\delta^{15}$N would be higher in the burned area than unburned area, especially in the organic soil, because live biomass in recently burned sites are still largely reduced, which would result in a larger inorganic nitrogen loss than its demand;

3) Plants in the burned sites would be more enriched in $^{15}$N than unburned forest because plants utilized the N resources that were $^{15}$N-enriched due to N losses.

## 2 Methods

### 2.1 Study area

Our study area was in Huzhong National Natural Reserve (51°17′42″ N to 51°56′31″ N, 122°42′14″ E to 123°18′05″ E), which is located in the Great Xing'an Mountains of northeastern China (Fig. 1). This

reserve encompassed 167,213 ha and experienced a terrestrial monsoon climate, characterized by a long and severe winder. The average annual temperature was -4.7 ℃ and mean annual precipitation was ca 500 mm. More than 60% of the annual precipitation fallen in the summer season from June to August (Liu *et al*. 2012; Zhou, 1991).

Dahurian larch (*Larix gmelinii*), a typical boreal conifer species, dominate the late successional

forests. Other tree species, such as pine (*Pinus sylvestris* var. *mongolica*), spruce (*Picea koraiensis*), birch (*Betula platyphylla*), two species of aspen (*Populus davidiana, Populus suaveolens*), willow (*Chosenia arbutifolia*), are interspersed with larch forest and have a small area of distribution (<2%) (Cai *et al*., 2013). Understory communities in the Great Xing'an mountains include *Vaccinium vitis-idaea*, *Ledum palustre*, *Carex schmidtii*, *Vaccinium uliginosum*, *Rhododendron dauricum*, and

*Rubus sachalinensis*. *Vaccinium vitis-idaea* and *Ledum palustre* are the mostly widely distributed understory species.

Wildfire is a major natural disturbance in this reserve. This area was characterized by frequent, low intensity surface fires mixed with infrequent stand-replacing fires. A sever wildfire burned 600 ha of

Huzhong National Natural Reserve on June 26th, 2010. This fire provided an ideal opportunity to study

the effects of fire on soil N dynamics in this ecosystem.

**2.2 Experimental design and field sampling**

In early June 2014, we randomly selected 12 plots (each 10 m ×10 m) in the burned area with six plots

at northern and southern slopes, respectively. In the unburned area, we also randomly set six plots (each

20 m × 20 m, Fig. 1) with three plots at each slope (northern and southern slope). To mitigate edge

effects, we located these plots at least 100 m away from the roads. In addition, each plot was 200 m

away from each other in order to minimize samples' spatial autocorrelation.

Soil samples were taken from two layers (organic layer and 0-20 cm mineral layer) at five random

locations within each plot and were composited. We also recorded the temperature of organic layer by

soil thermometer. On the same day the soil samples were collected, 7.5 g fresh organic soil passed

through 5 mm sieve and 30 g fresh mineral soil passed through 2 mm sieve were extracted by 75 ml 2

M KCl solution, shaking for 1 h at 160 r/m and then filtrated. The extracts were frozen and maintained

at -20 ℃ until later laboratory analysis. For each plant species, the foliage was sampled from at least 5

separate individuals within the same plot.

In early June 2015, we further collected plant and soil samples and separated them to different

components in the same plots. For plant tissues, foliage and branch were sampled from at least 5



separate individuals. Mosses were collected at five random locations within each plot and were

composited. Fine roots were separated from forest floor (Oa+e layer) and different mineral soil profiles

(0-10 cm, 10-20 cm). For soils, forest litter (Oi), forest floor (Oa+e layer) and three mineral soil profiles

(0-10 cm, 10-20 cm and 20-30 cm) samples were collected using the same method as the one used in

year 2014.

### 2.3 Laboratory treatment and chemical analysis

Subsamples of organic and mineral soils were air-dried, crushed and sieved through 5 mm and 2 mm

mesh, respectively, for chemical analysis. Soil pH was measured in $H_2O$ employing a soil: solution ratio

of 1:10. Soil samples were dried at 105 ℃ for 48 h to measure soil water content. Ammonium in the

extract was determined by the indophenol blue method followed by colorimetry, and $NO_3^-$ was

determined colorimely using the same autoanalyser in the form of $NO_2^-$ after reduction of $NO_3^-$ in a

Cd-Cu column followed by the reaction of $NO_2^-$ with N-1-napthylethylenediamine to produce a

chromophore (Rivas *et al.*, 2012).

     Plant and soil samples were dried at 60 ℃ to constant weight and ground into powder using a ball

mill and used to analyze $^{15}N$ natural abundance (expressed as $\delta^{15}N$), N and C concentrations by

elemental analyzer (vario MICRO cube; Elementar Analysensysteme GmbH, Hessen Hanau, Germany)

coupled to an IsoPrime100 continuous flow IRMS instrument. Calibrated glycine ($\delta^{15}N = 1.6‰$),



D-glutamic ($\delta^{15}N$ = -5.7‰), L-histidine ($\delta^{15}N$ = -7.6‰), and acetanilide ($\delta^{15}N$ = 1.4‰) were used as the internal standards. The $\delta^{15}N$ of the sample relative to the standard (atmospheric $N_2$) was expressed as the following:

$$\delta^{15}N = [(R_{sample}/R_{standard}) - 1] * 1000;$$

where $R_{sample}$ represents the isotope ratio ($^{15}N/^{14}N$) of sample and $R_{standard}$ is the $^{15}N/^{14}N$ for atmospheric $N_2$. The analytical precision for $\delta^{15}N$ was in general better than 0.2‰.

In order to examine the net N mineralization rate, we collected soil from the same plots in early Autumn of 2014 with the same sampling method. We used the soil samples collected in the late growing season for this purpose because we didn't have low-temperature sample transportation facilities during the first-round soil collection in early June. Net N mineralization rates were estimated using laboratory soil incubations. 7.5 g fresh organic soil passed through 5 mm sieve and 30 g fresh mineral soil passed through 2 mm sieve were put into a plastic cups with polyethylene film to minimize moisture evaporation and incubated at 20 ℃ for 1 week without light. Incubated soil mineral N ($NH_4^+$ and $NO_3^-$) was extracted and measured as above mentioned. Net N mineralization potentials were calculated as the difference between final and initial inorganic N ($NH_4^+$ + $NO_3^-$) concentrations divided by the number of incubation days. The expression "N mineralization potential" is used to designate soil samples that produced net amounts of inorganic N.

### 2.4 Statistical analysis

We used one-way analyses of variance (ANOVA) to test whether wildfire significantly affected soil N availability and examine the differences of $\delta^{15}N$ (‰) of foliage, organic soil and mineral soil in burned and unburned area. Significance level was set at a *P* value of 0.05 unless otherwise stated. Significant differences among treatment means of soil properties were analyzed using One- way ANOVA. Data were statistically analyzed in R (R Core Team, 2014).

### 3 Results

### 3.1 Basic soil properties

The alteration of soil basic properties in the burned area 4 years after the wildfire was mainly found in the organic soil, not in the mineral soil (Table 1). The soil water content (SWC), TN, TC, C:N were lower in the burned soil, but only the reduction of SWC and TC reached the significant level ($p <= 0.05$). Mean soil water content at the organic layer was reduced from 117.6% in the unburned sites to 41.2% in the burned sites. Mean TC at the organic layer was reduced from 29.2% to 9.2%. In contrast of those properties that were reduced after fire, pH and temperature were increased. The organic soil temperature was increased significantly from 2.9 °C in the unburned area to 10.0 °C in the burned area.

### 3.2 Soil inorganic nitrogen concentrations

Total inorganic N pools were greater in the organic soil than the mineral soil both in burned and

unburned sites (Fig. 2A). However, the significant increases in soil inorganic N concentrations in

response to wildfire were only observed in the mineral soil. Mean total inorganic N concentration in the

mineral soil was increased to 5.55 mg N kg$^{-1}$ in the burned sites from 2.22 mg N kg$^{-1}$ in the unburned

area. Compared to unburned sites, the amount of $NH_4^+$ was increased from 1.6 to 5.0 mg N kg$^{-1}$ in

mineral soil (Fig. 2B). $NO_3^-$ concentrations were consistently low in both organic and mineral soil, and

did not differ between burned and unburned area (Fig. 2C).

## 3.3 Nitrogen transformation rates

The response pattern of N transformation after the fire was similar to that of soil inorganic N

concentrations. Both net mineralization and ammonification rates were significantly increased to 0.056

mg N kg$^{-1}$d$^{-1}$ and 0.029 mg N kg$^{-1}$d$^{-1}$ in the organic soil of the burned area compared to the unburned

area of -0.653 mg N kg$^{-1}$d$^{-1}$ and -0.579 mg N kg$^{-1}$d$^{-1}$, respectively (Fig. 2D and Fig. 2E). In contrast,

ANOVA revealed no differences on both mineralization and ammonification rates in burned and

unburned mineral soil. There was no significant difference in net nitrification either in the organic or in

the mineral soil between burned and unburned soil (Fig. 2F).

## 3.4 Plant and soil δ$^{15}$N

Mean foliar $\delta^{15}N$ in the burned sites was 3.7‰ and was significantly higher compared to the mean foliar $\delta^{15}N$ in unburned site (-3.7‰) in the 4 years after fire (Fig. 3). The species occurring on both the burned and unburned sites such as *Vaccinium vitis-idaea*, *Ledum palustre* and *Deyeuxia angustifolia* had

significantly higher foliar $\delta^{15}N$ values in the burned area than the unburned area. The values for *Vaccinium vitis-idaea*, *Ledum palustre* and *Deyeuxia angustifolia* were 0.2‰, 2.6‰ and 1.8‰, respectively, in the burned area. In the unburned area, their corresponding values were -3.7‰, -3.4‰ and -2.4‰, respectively (Table 2). A significant difference of $\delta^{15}N$ between the burned (3.6‰) and unburned (1.3‰) area was also found in the organic soil (Fig. 3). However, there was no significant

difference in the mineral soil between burned (4.9‰) and unburned (4.8‰) area (Fig. 3).

The effects of wildfire on $\delta^{15}N$ were also detected in plants' aboveground parts in the burned area 5 years after the fire (Fig. 4). The mean foliar and branch $\delta^{15}N$ were 2.3‰ and 1.5‰, respectively, in the burned area, and were significantly ($p<0.001$) greater than in the unburned area (both -4.1‰). The moss $\delta^{15}N$ ranged from 0.9‰ to 1.7‰, and the mean was 0.7‰, which was significant ($p<0.001$) higher than

the moss collected in the unburned area (-4.1‰). Fine root $\delta^{15}N$ was significantly ($p<0.001$) increased from -0.9‰ to 0.7‰ after fire. As the various sub soil organic layers, the Oi was more depleted in $^{15}N$ (-3.6‰) in unburned area than in burned area (-2.4‰). The wildfire also significantly increased the $\delta^{15}N$ of Oa+e and 0-10 cm mineral soil, but had no significant effects on the deeper mineral soil layers.

**4 Discussions**

**4.1 The effect of fire on soil inorganic nitrogen concentrations and transformation rates**

We initially expected that the inorganic N concentrations and N mineralization rates in the burned area would have recovered to the pre-fire level 4 years after fire. However, our data didn't fully support this hypothesis. In contrast, our study showed that wildfire still had a strong effect on inorganic nitrogen concentrations and N mineralization rates. Specifically, fire-induced increases in $NH_4^+$ and TIN concentrations were observed in the mineral soil; and N mineralization and ammonification were increased significantly in the organic soil (Fig. 2). The observed increases of net mineralization rate and ammonification rate in the organic soil suggest that the TIN and $NH_4^+$ production rates might have exceeded consumption rates. However, there were no significant increases in TIN and $NH_4^+$ concentration between the burned and unburned area in the organic soil layer, which may be due to the increased plant uptake or N loss through gases and leaching. As to the mineral soil, although the net mineralization and ammonification in the burned area didn't exhibit an increase, the inorganic nitrogen concentrations were increased significantly, suggesting inorganic nitrogen infiltration from the organic soil to the mineral soil.

The significantly increased rates of net mineralization and net ammonification in organic soil after fire might be attributed to following three reasons: (1) increased available organic matter to microbes



(Dannenmann *et al*., 2011), as shown in our results that C:N ratio decreased from 27.5 to 20.8 (Table 1), may enhance microbial activities to decompose litter. (2) post-fire abiotic environments such as increased soil pH and temperature (Table 1) tend to be more suitable for microbial activities (Smithwick *et al*., 2005). Increased temperature might have played a key role in N transformation because decomposition rates may increase by 50% - 100% when soil temperatures increase $5\ ^{o}C – 10\ ^{o}C$ (Richter *et al*., 2000). In this study, organic soil temperature was increased (by $7.1\ ^{o}C$) significantly after fire, mainly due to combustion of thick organic layer (the thickness was decreased from 22.2 cm to 5.3 cm) and the removal of overstory tree, leading to more solar radiation reaching ground surface (Christensen and Muller, 1975). On the contrary, net nitrification rate remained unchanged after fire, despite increased soil $NH_4^+$ (Fig. 2). This could be because nitrifier population size may be too low after fire and it may take some time to increase (Turner *et al*., 2007). In addition, fire might have an adverse effect on nitrifying microbes in the study soil, as some previous studies have suggested nitrifiers are more susceptible to fire than other soil microbial groups (Hart *et al*., 2005).

Other studies have also observed increase in inorganic N after fire (Certini, 2005; Gomez-Rey and Gonzalez-Prieto, 2013; Koyama *et al*., 2012). Turner (2007) studied inorganic N pools and mineralization rates in the first 3 years after a stand-replacing wildfire in the Greater Yellowstone ecosystems and found that soil $NH_4^+$ concentration increased and followed by increases in soil $NO_3^-$, but

fire had a net negative influence on N mineralization due to microbe immobilization. Koyama *et al.* (2010) found soil $NO_3^-$ concentrations elevated in the 2 years after wildfire in the coniferous forests of central Idaho resulted from reduced microbial $NO_3^-$ uptake capacity, but $NH_4^+$ concentrations between the treatments were not significantly different. They also suggested that reduced available C was the key factor regulating soil N cycling after fire. On the contrary, Deluca and Sala (2006) showed recurrent, low-severing fire had a different effect on N in ponderosa pine forests. Post-fire soil total N concentrations and potential mineral N (PMN) rates decreased and the concentrations of $NH_4^+$ and $NO_3^-$ were not in line with the changes of total N pool and PMN rate. These studies collectively showed fire severity, time after fire, vegetation type and soil sampling depth may be responsible for the inconsistency of the reported findings (Wan *et al.*, 2001; Wang *et al.*, 2014).

## 4.2 The effect of fire on soil $\delta^{15}N$

Our results showed that $^{15}N$ natural abundance in organic soil was significantly higher in the burned area than unburned area (Fig. 3). These results are consistent with our expectation that soil $\delta^{15}N$ would be higher in the burned area than unburned area, especially in the organic soil. Similar results were reported after wildfire in other forest ecosystems (LeDuc *et al.*, 2013; Schafer and Mack, 2010). Combustion of the upper $\delta^{15}N$-depleted surface soil layer and enhanced nitrification are two widely-recognized mechanisms to explain the $^{15}N$ enrichment in organic soil (Hogberg, 1997; LeDuc *et*

*al*., 2013; Schafer and Mack, 2010).

We suggest, however, that fire-stimulated $NH_3$ volatilization is the primary soil N process resulting in $^{15}N$ enrichment of bulk soil in our study. A significant increase of $\delta^{15}N$ in soil was in line with the observation of higher net mineralization and net ammonification in burned organic soils (Figs. 2 and 3). High soil $NH_4^+$ pools in organic soil did not lead to an elevated net nitrification rate, suggesting a large

amount of $NH_4^+$ didn't transform to $NO_3^-$ through nitrification. Plant biomass in the burned area was 8.5 $Mg\ ha^{-1}$, which was much less than unburned area (76.9 $Mg\ ha^{-1}$, *unpublished data*). So a large amount of $NH_4^+$ was lost through volatilization since few plants were present after fire and plant biomass had not recovered to the pre-fire level. $NH_3$ volatilization is associated with strong fractionation against $^{15}N$ and higher gasses losses of $^{15}N$-depleted $NH_3$, and leads to the remaining soil $NH_4^+$ to be enriched in

$^{15}N$ (Hobbie and Ouimette, 2009). Furthermore, higher soil temperature as observed in the present study would also facilitate $NH_3$ volatilization. Our suggestion is in agreement with the results of Raison (1979), who also confirmed $NH_3$ volatilization from fire can be significant. Therefore, fire-stimulated $NH_3$ volatilization associated with strong isotopic fractionation and subsequent export of $^{15}N$-deplted $NO_3^-$ is considered as being responsible for $^{15}N$ enrichment of bulk soil.

Fang *et al.* (2015) reported that denitrification was a significant N loss pathway and could account for 48 to 86% total $NO_3^-$ loss in forest ecosystems. Denitrification might play a critical role in higher soil

$\delta^{15}$N even though we didn't measure this N process directly. This speculation is supported by our results. On one hand, less N competitor, such as lower plant and microbial biomass after fire, would result in lower N need and more $NO_3^-$ loss through denitrification. On the other hand, the significant increase in

net nitrification rates was not observed in laboratory experiment is also likely due to an enhanced denitrification, which is associated with strong fractionation against $^{15}$N and higher gasses losses of $^{15}$N -depleted $N_2$ or $N_2O$, remaining soil $NO_3^-$ to be enriched in $^{15}$N (Hobbie and Ouimette, 2009; Robinson, 2001).

Combustion of surface soil layer could cause the upper soil to be enriched in $^{15}$N since high,

sustained fire temperatures cause a greater loss of $^{14}$N compared to $^{15}$N (Huber *et al*., 2013; Schafer and Mack, 2010). However, this increased $\delta^{15}$N were only observed in organic layer and 0-10cm mineral soil due to the thick organic layer insulate underlying mineral soil from heating and limit downward conduction of heat (Smithwick, *et al*., 2005). Thus low fire temperature in deeper mineral soil could explain the lack of an effect of fire on mineral soil $\delta^{15}$N.

Finally, the $^{15}$N enriched litter return could cause the upper soil to become enriched in $^{15}$N after fire. Plant tissues fallen onto the surface soil, resulting in litter with a similar value of $\delta^{15}$N. In mature larch boreal forest where N is limited, the $^{15}$N-depleted leaf could lead to a lower $\delta^{15}$N in litter. In addition, Oi was composed of a large number of $^{15}$N-depleted coarse woody debris and a small number of recently



added litter with a higher $\delta^{15}N$ value, which contribute to a relative lower $\delta^{15}N$ in Oi.

**4.3 The effect of fire on plant $\delta^{15}N$**

Foliar $\delta^{15}N$ values were significantly increased in the burned area, which supports our initial expectation that plant $\delta^{15}N$ in the burned forest would be enriched in $^{15}N$. Three complementary processes are likely responsible for this $^{15}N$ enrichment. First, fire consumed the $^{15}N$-depleted surface layers of litter, forcing plants to take up the N from deeper horizons which are more enriched $^{15}N$ than

the surface soil (Hogberg, 1997; Sah *et al*., 2006). This assumption is supported by our field experiment in 2015. We found the root was reduced largely in the organic layer and fine roots were mainly distributed in the 0-20 cm mineral soil in the burned area; while in the unburned area, fine roots were mainly distributed in the organic soil layer. Secondly, part of $^{15}N$-enriched $NH_4^+$ and $NO_3^-$ infiltrated into the deeper mineral soil with rainfall from the organic layer, which leads to the remaining soil N

pool to be enriched in $^{15}N$ and further expressed in the values of $\delta^{15}N$ in plant that utilized these N pools for their N demand. Thirdly, increased N availability could lead to a lower dependence of plant N nutrition upon mycorrhizal fungi, which provide their host plants with $^{15}N$-depleted N relative to the soil N sources (Craine *et al*., 2009; Hobbie *et al*., 2008). Therefore, plant foliar $\delta^{15}N$ values tend to be higher in the burned area with higher N availability and a lower dependence on mycorrhizal fungi.

*Vaccinium vitis-idaea, Ledum palustre* were species occurring in both burned and unburned area and

they are associated with ERM mycorrhizal fungi (Michelsen *et al*., 1998). Because there were lower inorganic concentrations in unburned soil than burned soil, in order to acquire enough inorganic nitrogen, those plants in the unburned area had to depend on mycorrhizal fungi transferring $^{15}$N-depleted N to them.

Furthermore, plant $\delta^{15}$N could provide insights into the ecosystem N openness, with higher values indicating a more open N cycle (Hietz *et al*., 2011; Pardo *et al*., 2006). The unburned area (mature larch boreal forest) is a typical N limited ecosystem and has a negative foliar $\delta^{15}$N (-3.7‰). The average foliar $\delta^{15}$N values were 3.7‰ and 2.4‰, respectively, in the 4 and 5 years after fire (Fig. 4), suggesting this ecosystem has shifted from N limited to N open. Natural ecosystem N openness is often resulted

from N loss and the more N lost, the more open the ecosystem is. Therefore, we can expect that an ecosystem will become more N open if available N supply exceeds demand, resulting in an increase in N loss despite at a low available N level. Our results show that the concentration of $NH_4^+$, the predominant inorganic N form in this boreal forest (Cheng *et al*., 2010), recovered to the pre-fire level in the 4 years after fire. Whereas, the live biomass in the burned area was greatly reduced after fire and

lower than that of the unburned area. As a consequence, a large amount of available N would exceed the demand in the burned area and lost through ammonium volatilization and denitrification, discriminating against $^{15}$N, leading to a higher $\delta^{15}$N in soil and plant. Our results suggested that natural abundance of

$^{15}$N is a more comprehensive index for predicting N openness after wildfires than available nitrogen concentration.

**5 Conclusions**

In this study we demonstrated that wildfire had a profound influence on N cycles in the boreal forests of the Great Xing'an Mountains. The ecosystem N cycle was still open in 4 and 5 years after fire. However, the wildfire effects were limited in organic layer and 0-10cm mineral soil. The fire-induced increases in net mineralization rate and net ammonification rate were only exhibited in the organic soil, not in the mineral soil. The increased organic layer temperature, decreased moisture and C:N could be the primary mechanism determining inorganic N transformation rates. Wildfire could cause severe N losses through combustion and $NH_3$ volatilization, which may be the dominant N processes contributing to increased $\delta^{15}$N values in plant and soil. The $\delta^{15}$N of plant and soil could be considered as a comprehensive indicator for explore the responses of N processes to wildfire in forest ecosystems.

**Acknowledgement**

This research was supported by the National Natural Science Foundation of China (41222004, 31270511, 31422009, 41301200) and State Key Laboratory of Forest and Soil Ecology, Institute of Applied Ecology, the Chinese Academy of Sciences (No. LFSE 2013-13). We acknowledge Jiaxing Zu and Yue

Yu for their assistance in the field work. We also thank staffs in Huzhong National Natural Reserve for

their supports in the field sampling.

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

**Figure captions**

Figure 1. Research location in the Huzhong Natural Reserve (HNR), China. A large wildfire burned almost 600 ha mature larch forest within the HNR in the summer of 2010. The red boundary represents the burned area. Unburned area is chosen in the nearby burned area as control. The black triangles represent burned plots, the yellow circles represent unburned plots.

Figure 2. Soil inorganic N concentrations (TIN, $NO_3^-$-N and $NH_4^+$-N) and N transformation rates (net mineralization, ammonification and nitrification) of two soil layers. Shown are mean $\pm$ one standard error. Different letters indicate significant difference between burned and unburned plots.

Figure 3. $\delta^{15}N$ values (‰) of foliage, organic soil and mineral soils. Shown are mean $\pm$ one standard errors. Different letters indicate significant difference between burned and unburned plots.

Figure 4. $\delta^{15}N$ (‰) for plant and soil in unburned and burned systems 5 years after wildfire. Solid red circles represent burned plots, solid black squares represent unburned plots, respectively. One asterisk indicate significant difference among forests at $p < 0.05$, two asterisks indicate significant difference among forests at $p < 0.01$, three asterisks indicate significant difference among forests at $p < 0.001$.

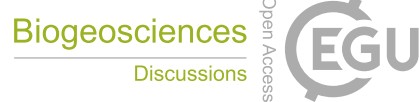



Table 1. Basic soil properties at two soil layer in unburned (n=12) and burned area (n=24). Values presented are means with the standard error in parentheses. Means

in a column that have the same letter are not significantly different at alpha level is 0.05 (ANOVA, $p \leq 0.05$).

| Layer | SWC (%) | | PH | | TN (%) | | TC (%) | | C:N | | T(℃) | |
|---|---|---|---|---|---|---|---|---|---|---|---|---|
| | Unburned | Burned | Unburned | Burned | Unburned | Burned | Unburned | Burned | Unburned | Burned | Unburned | Burned |
| Organic layer | 117.6 (32.0)a | 41.2 (18.6)b | 4.4 (0.5)a | 5.2 (0.5)a | 0.8 (0.5)a | 0.4 (0.2)a | 29.2 (10.2)a | 9.2 (4.0)b | 27.5 (5.0)a | 20.8 (5.4)a | 2.9 (1.1)a | 10.0 (2.5)b |
| Mineral layer | 36.2 (8.0)a | 35.2 (7.7)a | 5.2 (0.4)a | 5.3 (0.3)a | 0.2 (0.0)a | 0.2 (0.0)a | 4.3 (1.1)a | 3.1 (0.5)a | 24.1 (4.1)a | 20.3 (2.4)a | NA | NA |





Table 2. Foliar stable N isotope ratio ($\delta^{15}$N), N concentration, C concentration and C:N ratios for each sampled species in burned and unburned area.

| Site location | Species | $\delta^{15}$N (‰) | N conc.(%) | C conc.(%) | C:N ratio |
|---|---|---|---|---|---|
| Burned polts | *Vaccinium vitis-idaea* | 0.2 | 1.4 | 50.1 | 36.1 |
| | *Ledum palustre* | 2.6 | 2.1 | 51.8 | 25.2 |
| | *Deyeuxia angustifolia* | 1.8 | 2.6 | 43.7 | 17.1 |
| | *Carex schmidtii* | 3.4 | 2.1 | 42.9 | 21.4 |
| | *Chamerion angustifolium* | 5.2 | 3.9 | 46 | 12 |
| | *Betula platyphylla* | 2.4 | 3.1 | 48 | 15.7 |
| | *Rubus sachalinensis* | 3.4 | 2.8 | 44.5 | 16 |
| | Mean±SE | 3.7±1.9 | 2.9±0.9 | 45.5±2.6 | 17.3±5.5 |
| Unburned area | *Vaccinium vitis-idaea* | -3.7 | 1.2 | 49.4 | 40.9 |
| | *Ledum palustre* | -3.4 | 2 | 51.5 | 26.3 |
| | *Deyeuxia angustifolia* | -2.4 | 2.3 | 42.7 | 18.3 |
| | *Pinus pumila* | -4.2 | 1.3 | 49.3 | 39.8 |
| | *Larix gmelini* | -4.6 | 1.6 | 48.2 | 31.2 |
| | *Rhododendron dauricum* | -2.9 | 2.1 | 48 | 22.6 |
| | Mean±SE | -3.7±1.3 | 1.7±0.5 | 48.7±2.0 | 31.5±9.0 |





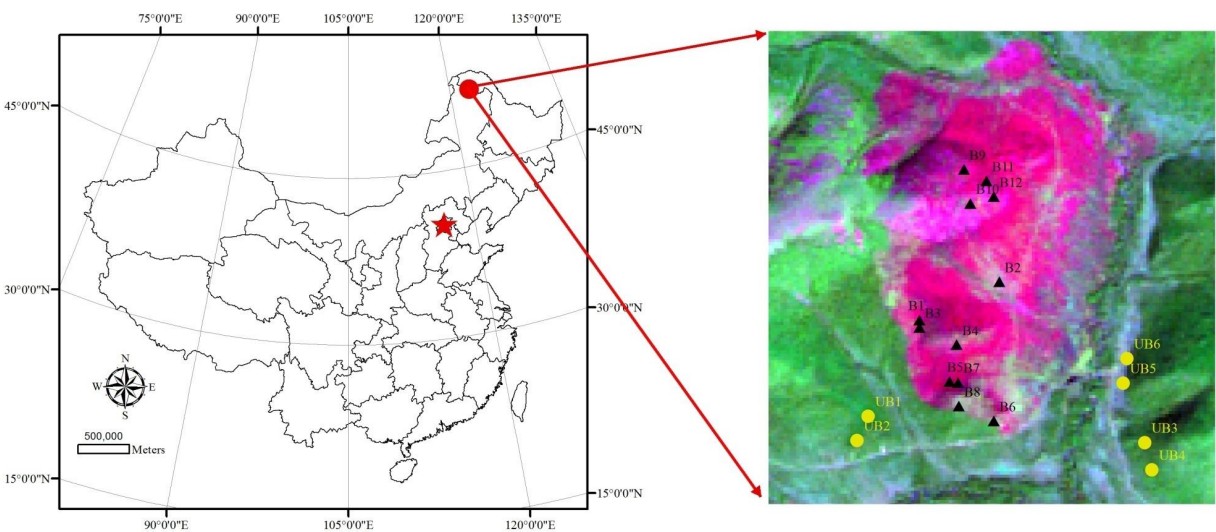

Figure 1.Research location in the Huzhong Natural Reserve (HNR), China. A large wildfire burned almost 600 ha mature larch forest within the HNR in the summer of 2010. The red area represents the burned area. Unburned area is chosen in the nearby burned area as control. The black triangles represent burned plots, the yellow circles represent unburned plots.





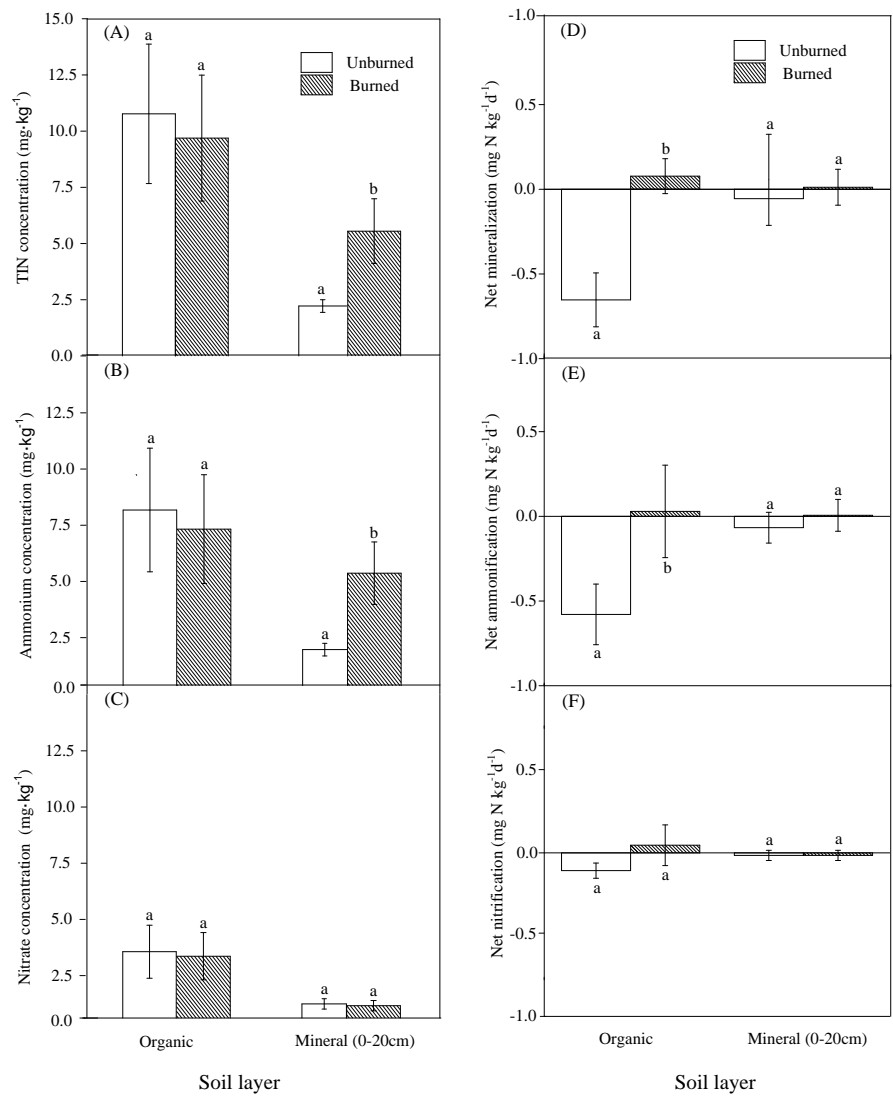

Figure 2. Soil inorganic N concentrations (TIN, $NO_3^-$-N and $NH_4^+$-N) and N transformation rates (net mineralization, ammonification and nitrification) of two soil layers. Shown are mean ±one standard error. Different letters indicate significant difference between burned and unburned plots.





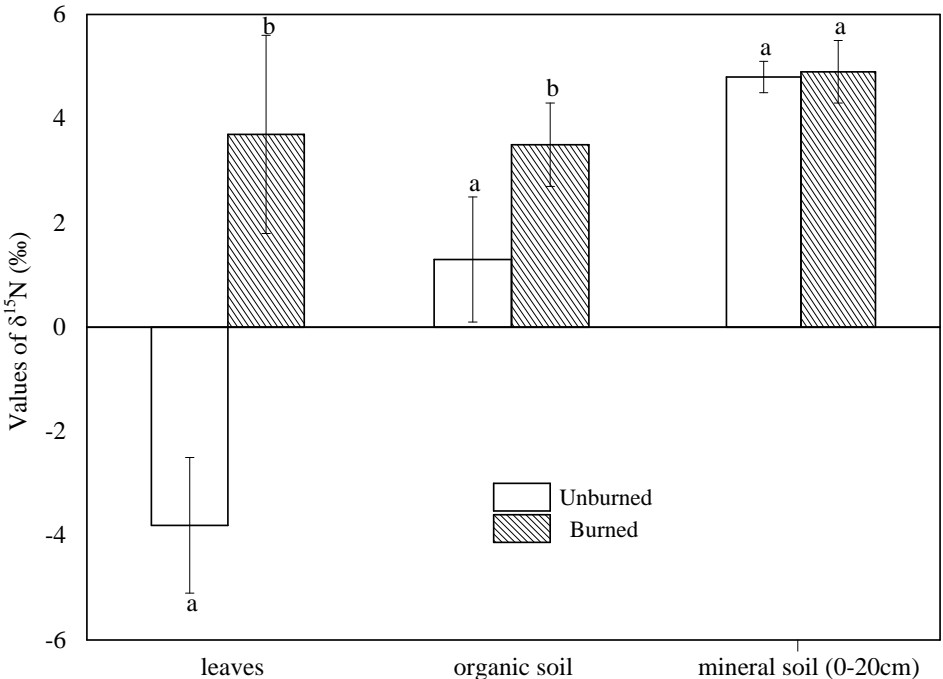

Figure 3. $\delta^{15}$N values (‰) of foliage, organic soil and mineral soils. Shown are mean ± one standard

errors. Different letters indicate significant difference between burned and unburned sites.





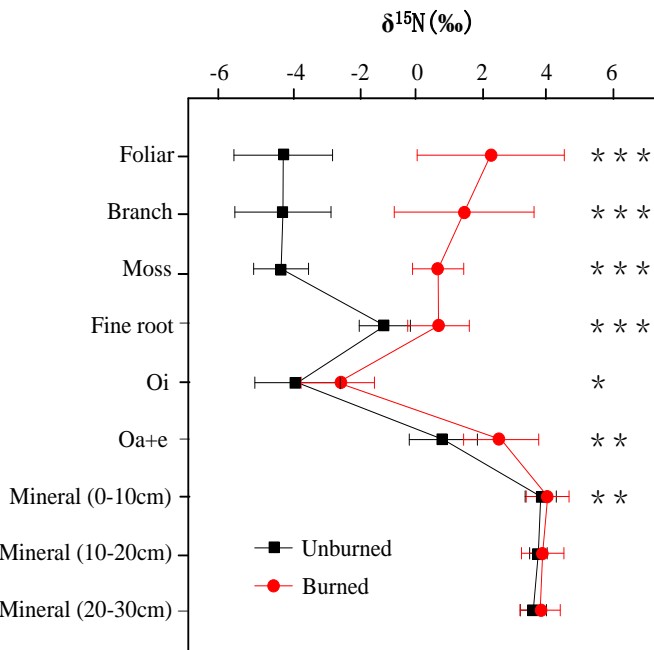

Figure 4. $\delta^{15}N$ (‰) for plant and soil in unburned and burned systems 5 years after wildfire. Solid red circles represent burned plots, solid black squares represent unburned plots, respectively. One asterisk indicate significant difference among forests at $p < 0.05$, two asterisks indicate significant difference among forests at $p < 0.01$, three asterisks indicate significant difference among forests at $p < 0.001$.