# Peer review of "Wildfire effects on ecosystem nitrogen cycling in a Chinese boreal larch forest, revealed by $^{15}\text{N}$ natural abundance"

_Biogeosciences, 2016_

## Referee Comment (RC1) · Anonymous Referee #1 · 30 Apr 2016

Manuscript ID: bg-2016-91

This manuscript examined the wildfire effect on nitrogen cycles in a boreal forest in northeastern China using 15N natural abundance. The results showed higher TIN and ammonium concentrations in the mineral soil and higher net mineralization rates in the organic soil in the sites which was burned 4 years ago, compared to the control sites. The N isotopic ratios of organic soil and plant leaves were greater in the burned sites than in the control. The authors propose that the environmental modification caused by wildfire affected the soil N dynamics and ammonium volatilization was the main driver of the N loss, which lead to 15N-enrichment in the burned sites. As the authors describe in Introduction, the increased frequency of wildfire and its consequences on

N cycles are a global environmental issue. I think that this manuscript is concisely well written. However, there are several issues as mentioned below to be addressed before recommendation for publication in Biogeosciences can be made.

General comments: The pattern that 15N values of plant and soil increase after wildfire and return to the pre-fire values has widely been observed in many terrestrial ecosystems (Szpak, 2014). This study provides new data on the isotopic pattern from a boreal forest in China. In this manuscript, the authors conclude that ammonium volatilization is the main driver of the increase of 15N (Line28, Lines 282-284). Although I also think ammonium volatilization is one of possible explanations about the increase of 15N, there seems not enough evidence supporting this conclusion. The authors argued that difference in ammonium and nitrate pools and N mineralization rates between burned and unburned sites could be due to ammonium volatilization. However, the present results were based on only one-time estimation, and thus they seem not convincing enough to suggest the occurrence of greater ammonium volatilization in the burned sites. As the authors discussed, N loss due to nitrate leaching could also have caused the enrichment in soil 15N. Further, greater dependence of plants on deeper soil N, or less dependence on N derived from mycorrhizal fungi in the burned sites might also have increased the 15N. These 15N-enrichment mechanisms appear to be equally plausible at this stage. To explore the enrichment mechanisms, it would be necessary to investigate N budget of the studied forests, 15N values of DIN and TN, root distribution, and infection rates of mycorrhizal fungi in the roots. Unfortunately, however, these data were not presented in this manuscript. As such, the authors would need to modify Discussion section to discuss these explanations more carefully about the increase of 15N.

The discussion about openness of ecosystem N cycle (Line 325) could be made in conjunction with the earlier paragraphs about ammonium volatilization and nitrate leaching, because these two processes are the main N loss pathways when N cycle become open.

The authors used average 15N values of all plant species for the comparison between burned and unburned sites (Table 2). However, this comparison would need to be performed for each plant species, because different plant species have distinct niches for N acquisition, which are reflected in variation in 15N among plant species (Table 2).

Reference Szpak, P., 2014. Complexities of nitrogen isotope biogeochemistry in plant-soil systems: implications for the study of ancient agricultural and animal management practices. Front Plant Sci 5, 288.

Minor comments: L79: A reference would be needed. L128: Please explain in more details how the soil temperature was measured. L134: Please add more explanations about the sampling (e.g., tree species). L136: Please describe the species name of the moss. L170: total nitrogen (TN), total carbon (TC) L181: It would be better to avoid the expression "XX was reduced (or increased) after fire" throughout this manuscript, because this study did not examine the pre- and post-fire effects in the same sites, but compared the soil properties between burned and unburned sites. L259: Please add more explanation about "potential mineral N rates". L281: References would be needed for the effect of soil temperature on NH4 volatilization. L289-293: Please clarify this sentence.

---

## Referee Comment (RC2) · Anonymous Referee #2 · 15 May 2016

This manuscript seeks to determine the effects of fire, 4-5 years after burning, on soil and plant N cycling using a combination of soil and litter available inorganic N pools and fluxes and soil and plant 15N. This manuscript has several major issues that would need to be addressed before it could be published in Biogeosciences. Some of these concerns could, in theory, be addressed with major revision, but it would require another review as the manuscript would be completely changed. I hope my suggestions below are useful to the authors.

This is a largely observational study that also attempts to identify mechanisms. Though the two concepts are not exclusive, they can be challenging to reconcile, and the challenge is apparent here. The observation is that, 4-5 years after fire, there are higher

[Figure]

N fluxes in organic soil, higher N pools in mineral soil, and higher 15N values in plants and soil in burned forest than unburned forest. The "mechanisms" that explain these observations are derived from patterns within the observations, but there are two problems associated with this extrapolation:

1) I was concerned by the fact that the mineralization results are compared to the TIN results, despite the fact that they were taken in different seasons. specifically, the TIN samples were collected before (June) the wettest part of the year (June to August; Line 107), whereas the N mineralization samples were collected after the wet season (Autumn, but not specified). Though it is well known that the size and direction of N pools and fluxes can change seasonally, these data are compared to each other as though they represent the same N. For example, the authors posit that the "high soil NH4+ pools did not lead to an elevated net nitrification rate" (Line 274), but the NH4+ pool and the net nitrification rate were collected months apart and likely had little bearing on each other. These two time points should not be compared this way.

2) As a result of the previous comment, the "mechanisms" that could explain these patterns of N cycling are difficult to establish, at best. The authors state that "a large amount of NH4+ was lost through volatilization" (Lines 276-277). However, there is no empirical evidence for this, and no way in which to rule out other mechanisms that explain differences in 15N isotopic signatures between the sites, such as denitrification or combustion (the authors addressed these, but emphasized volatilization as the proximate mechanism controlling 15N).

Throughout, from the title through the discussion, there is a strong emphasis placed on the revelation that 15N represented an "open" N cycle. For example, in the introduction, it is stated that "d15N could provide us a promising and comprehensive tool to detect the effect of wildfire on N cycling" (Lines 57-58). This makes it sound like establishing this is an objective of the paper, but this is a well-established use of these measurements (see Martinelli et al. 1999). By contrast, the important result related to the stated objectives would seem to be that fire leaves an open N cycle for several

years. Likely, this problem could be addressed by changing verbiage.

There was a paucity of information about the fire: it was described as severe (line 118), but little other information was provided than unpublished results, not explained in the methods, that there was a tenfold loss of aboveground biomass (Lines 275-276). I wonder why the authors would expect to see a recovery of the N cycle given such substantial fire-related effects persist? This was not clearly articulated in the manuscript- what was the underlying rationale for this work, other than the lack of data on N cycling from Chinese larch forest?

I was also surprised that there was no mention of the role of cation exchange capacity in N cycling. While fluxes of N were affected by fire in the litter, N pools were affected in the soil. This likely represents the simple fact that a changed physical environment, combined with a reduction in plant uptake, means greater microbial processing of N in the organic layer and greater sorption of N in the mineral soil. This wasn't clearly articulated despite the fact that several lines were dedicated to changes in temperature (Lines 239-244).

Some specific comments: There were a number of typos and awkward phrases that would need to be cleared up before publication (for example: "winder" instead of winter (Line 106), "sever" instead of severe (Line 118), "colorimely" instead of colorimetrically (Line 146), "filtrated" instead of filtered (Line 131), "burned polts" instead of burned plots (Table 2) – not a complete list).

There are better citations for studies suggesting N limits high latitude terrestrial ecosystems than Popova et al. 2013 and Stark & Hart 1997 (Lines 34-35). Consider Vitousek & Howarth 1991, Lebauer and Treseder 2008, Elser et al. 2207, and Harpole et al. 2011).

How is Figure 3 different from Figure 4? Similarly, how are the values provided in the paragraph beginning on Line 202 different from the values described in the paragraph beginning on Line 211? Both involve foliar d15N, for example. Not clear.

How was the branch, moss and fine root 15N relevant to the objectives? Seems like these data, especially the moss, are ancillary and it was not clearly stated that there was a goal to observe vertical profiles of d15N signatures from tree top to 30 cm mineral soil.

Figure 1 didn't come out very well, and wasn't very helpful. The caption makes it sound like the unburned area is in the burned area: "Unburned area is chosen in the nearby burned area as control."

References: Martinelli, L. A., M. C. Piccolo, A. R. Townsend, P. M. Vitousek, E. Cuevas, W. McDowell, G. P. Robertson, O. C. Santos, and K. Treseder (1999) Nitrogen stable isotopic composition of leaves and soil: Tropical versus temperate forests. Biogeochemistry, 46, 45–65.

Lebauer, D. S., and K. K. Treseder (2008) Nitrogen limitation of net primary productivity in terrestrial ecosystems is globally distributed. Ecology 89:371–379.

Harpole, W. S., et al. (2011) Nutrient co-limitation of primary producer communities. Ecology Letters, 14, 852–862.

Elser, J. J., et al. (2007) Global analysis of nitrogen and phosphorus limitation of primary producers in freshwater, marine, and terrestrial ecosystems. Ecology Letters, 10, 1135–1142.

Vitousek, P. M., and R. W. Howarth (1991) Nitrogen limitation on land and in the sea: how can it occur? Biogeochemistry, 13, 87–115.

---

## Author Comment (AC1) · 27 Jun 2016

We appreciate referee #1's positive assessment of our manuscript and the constructive comments. Replies to the raised points are provided below:

Reviewer 1's general comments: "General comments: The pattern that 15N values of plant and soil increase after wildfire and return to the pre-fire values has widely been observed in many terrestrial ecosystems (Szpak, 2014). This study provides new data on the isotopic pattern from a boreal forest in China. In this manuscript, the authors conclude that ammonium volatilization is the main driver of the increase of 15N (Line28, Lines 282-284). Although I also think ammonium volatilization is one of possible explanations about the increase of 15N, there seems not enough evidence supporting this

[Figure]

conclusion. The authors argued that difference in ammonium and nitrate pools and N mineralization rates between burned and unburned sites could be due to ammonium volatilization. However, the present results were based on only one-time estimation, and thus they seem not convincing enough to suggest the occurrence of greater ammonium volatilization in the burned sites. As the authors discussed, N loss due to nitrate leaching could also have caused the enrichment in soil 15N. Further, greater dependence of plants on deeper soil N, or less dependence on N derived from mycorrhizal fungi in the burned sites might also have increased the 15N. These 15N-enrichment mechanisms appear to be equally plausible at this stage. To explore the enrichment mechanisms, it would be necessary to investigate N budget of the studied forests, 15N values of DIN and TN, root distribution, and infection rates of mycorrhizal fungi in the roots. Unfortunately, however, these data were not presented in this manuscript. As such, the authors would need to modify Discussion section to discuss these explanations more carefully about the increase of 15N."

→Authors' response: Thanks for your constructive comments and helpful suggestions. We acknowledged that there are multiple mechanisms (e.g., NH3 volatilization, combustion, litter return, denitrification) that can contribute to the observed 15N enrichment in the soil. We no longer speculate that NH3 volatilization is the main driver of the increase of 15N. Instead, we provided a suite of mechanisms that might be equally plausible in explaining this observed pattern. Please see Lines 27-28, Lines 311-313 and Lines 372-374 for details on how we revised the discussion and conclusion to reflect this point. To address the reviewer's concern regarding different time points when comparing ammonification rate and ammonium concentration. We provided new data about inorganic nitrogen concentrations in August to ensure the comparison of these two variables were derived from the samples collected at the same time (Fig. 2D-F). We showed the ammonification rate of the organic soil in the burned area was higher than that in the unburned area. To the contrary, the ammonium concentration was significantly lower than that in the unburned area. Such reduction of ammonium could be very likely due to NH3 volatilization, although several other mechanisms such as surface run-off and filtration to mineral soil might also contribute to this observed pattern. We have revised the manuscript accordingly to illustrate this point. Please turn to the attached copy of the revised ms for details in Lines 215-220 and Lines 302-313.

Reviewer 1's comments: The discussion about openness of ecosystem N cycle (Line 325) could be made in conjunction with the earlier paragraphs about ammonium volatilization and nitrate leaching, because these two processes are the main N loss pathways when N cycle become open.

→Authors' response: We accepted the referee #1's comments and modified the discussion section. Please turn to the attached copy of the revised ms for details in Lines 311-313 and Lines 328-335 in 4.2.

Reviewer 1's comments: The authors used average 15N values of all plant species for the comparison between burned and unburned sites (Table 2). However, this comparison would need to be performed for each plant species, because different plant species have distinct niches for N acquisition, which are reflected in variation in 15N among plant species (Table 2).

→Authors' response: we completely agree with the referee #1 in that "different plant species have distinct niches for N acquisition, which are reflected in variation in 15N among plant species". We not only used Figure 4 to show average 15N value of all plant species in burned area was significantly higher than that in the unburned area, but also used Table 2 to show the differences in the foliar $\delta$15N in burned and unburned area for each specific plant species. Some plant species, such as Ledum, Vaccinium and Deyeuxia, can be observed in both burned and unburned area, whereas their foliar $\delta$15N in burned area were significantly higher than those in the unburned area. Therefore, Table 2 revealed that the differences of foliar $\delta$15N in burned and unburned area were resulted from differences in fire history.

Reviewer 1's comments: Minor comments: 1) L79: A reference would be needed. 2) L128: Please explain in more details how the soil temperature was measured. 3) L134:

Please add more explanations about the sampling (e.g., tree species). 4) L136: Please describe the species name of the moss. 5) L170: total nitrogen (TN), total carbon (TC) 6) L181: It would be better to avoid the expression "XX was reduced (or increased) after fire" throughout this manuscript, because this study did not examine the pre- and post-fire effects in the same sites, but compared the soil properties between burned and unburned sites. 7) L259: Please add more explanation about "potential mineral N rates". 8) L281: References would be needed for the effect of soil temperature on NH3 volatilization.9) L289-293: Please clarify this sentence.

→Authors' response: We thank the referee #1 for the thorough technical comments. These modifications are: 1) – We added the reference (Xu et al., 1997) in the L84; 2) – We added such information of soil temperature measure as the following "We also recorded the temperature of organic layer by soil thermometer at the soil depth of 5 cm (whenever applicable). The soil temperature was measured between 10am and 4pm. To account for the inherent hourly and daily temperature variations, we also measured soil temperatures at two fixed places at the hourly basis and used them as the baseline temperature data to correct such sources of uncertainty. The corrected values would be used to compare the difference in mean soil temperature between burned and unburned areas". 3) – We added the information of tree species both in burned and unburned area in as following: "In the unburned area, the dominant overstory species is Larix gmelinii, and the dominant understory species include Vaccinium vitis-idaea, Ledum palustre, Rhododendron dauricum, and Pinus pumila. In the burned area, the dominant species include seedlings of Larix gemlinii and some shrubs and herbs, such as Vaccinium vitis-idaea, Ledum palustre, Carex schmidtii and Rubus sachalinensis". 4) – We added the species name of moss as following: Different moss species were observed in unburned and burned area, Hypnum spp. was observed in the unburned area, whereas Polytrichum piliferum was the common moss species in the burned area. 5) – We added the "total nitrogen and total carbon" in L197; 6) – We accepted the referee #1's suggestion and replaced "XX was reduced (or increased) after fire" with the expression of "XX was higher (or lower) in burned area than that in unburned

area" throughout the manuscript; 7) – We added the information about "potential mineral N rates" by "using the 14-day anaerobic incubation procedure"; 8) – We added the reference (Nelson and Conrad, 1982) about the effect of soil temperature on NH3 volatilization; 9) – We changed the sentence to "On the other hand, the lack of increase in net nitrification in the burned organic soil resulted from 7-day laboratory incubation might be due to an enhanced denitrification, which is associated with strong fractionation against 15N and higher gaseous losses of 15N-depleted N2 or N2O, remaining soil NO3- to be enriched in 15N (Hobbie and Ouimette, 2009; Robinson, 2001)"

Please also note the supplement to this comment:
http://www.biogeosciences-discuss.net/bg-2016-91/bg-2016-91-AC1-supplement.pdf

———————————————————————

[Figure]

**Supplement:**

**Wildfire effects on ecosystem nitrogen cycling in a Chinese boreal larch forest, revealed by $^{15}$N natural abundance**

Weili Liu[1,2], Lin Qi[1], Yunting Fang[1], Jian Yang[1,3]*

[1]Key Laboratory of Forest and Soil Ecology, Institute of Applied Ecology, Chinese Academy of

Sciences, Shenyang 110164, P. R. China

[2]University of Chinese Academy of Sciences, Beijing 100049, P. R. China

[3]Department of Forestry, University of Kentucky, Lexington, KY, 40546, USA

*Correspondence to: J. Yang (jian.yang@uky.edu)

**Abstract.** Wildfire is reported to exert strong influences on N cycling, particularly during the early succession period immediately after burning (i.e., < 1 year). Previous studies have mainly focused on wildfires influences on inorganic N concentrations and N mineralization rates; but plant and soil $^{15}$N natural abundance (expressed by $\delta^{15}$N), as a spatial-temporal integrator of ecosystem N cycling, could provide a more comprehensive understanding of wildfire on various N cycling processes at a relatively broader time scale. In this study, we attempted to evaluate legacy effects of wildfire on nitrogen cycling using $\delta^{15}$N in a boreal forest of northeastern China, which is an important yet understudied ecosystem. We measured inorganic N concentrations ($NH_4^+$ and $NO_3^-$) and net N transformation rates (net ammonification, net nitrification, and net mineralization) of organic and mineral soil 4 years after a

wildfire and compared with unburned area. We also measured $\delta^{15}N$ of plant and soil samples in 4 and 5 years after the fire. We found that even 4 years after burning, net mineralization and net ammonification in the organic soil were still higher than those in the unburned area. $NH_4^+$ and total inorganic N (TIN) concentrations in the organic soil of the burned area did not significantly differ from those of the unburned area. Organic soil and foliar $\delta^{15}N$ were significantly higher (by 2.2‰ and 7.4‰, respectively) in the burned area than those in the unburned area. Five years after fire, $\delta^{15}N$ of plant tissues such as foliar, branch, fine roots and moss in the burned area were significantly greater (by 1.7‰ to 6.4‰) than that in unburned area. $\delta^{15}N$ of Oi, Oa+e and 0-10 cm mineral soil were also significantly higher in the burned area than unburned area, but showed no significant difference in deeper layer of mineral soil. The observed soil $^{15}N$ enrichment might be attributed to various mechanisms such as $NH_3$ volatilization, combustion, litter return, and denitrification. Greater dependence of plant on deeper soil N and less dependence on mycorrhizal fungi in the burned area might also have contributed to the increase of the $^{15}N$ in plant and soil. Such $^{15}N$ enrichments in soil and plant suggest that N cycling could remain openness years after fire disturbance has occurred, with N supply exceeding demand, leading to a great amount of nitrogen loss from the system for a relatively long time.

[revised manuscript text omitted]

120      The historical fire regime in this region is described as frequent, surface fires mixed with infrequent,

stand-replacing crown fires, with fire-free interval ranged from 30 to 120 years (Xu *et al*., 1997; Liu *et*

*al*., 2012). However, climate change, forest management and human activities have altered fire regimes

in this region (Jackson *et al*., 1997; Wang *et al*., 2007). Although the dominant tree species Dahurian

larch is regarded as a fire-tolerant species with thick bark near the stem bottom, its post-fire mortality

125      rate is still high, mainly due to a horizontal shallow-distributed root system (Fang *et al*., 2015;

Vijayakumar *et al*., 2016). A stand-replacing wildfire,which was ignited by lighting, burned 600 ha of

Huzhong National Natural Reserve on June 26th, 2010. This fire provided an ideal opportunity to study

the effects of fire on soil N dynamics in this ecosystem.

**2.2 Experimental design and field sampling**

130   In early June 2014, we randomly selected 12 plots (each 10 m ×10 m) in the burned area with six plots

at northern and southern slopes, respectively. In the unburned area, we also randomly set six plots (each

20 m × 20 m, Fig. 1) with three plots at each slope (northern and southern slope). To mitigate edge

effects, we located these plots at least 100 m away from the roads. In addition, each plot was 200 m

away from each other in order to minimize samples' spatial autocorrelation.

135      Soil samples were taken from two layers (organic layer and 0-20 cm mineral layer) at five random

locations within each plot and were composited. We also recorded the temperature of organic layer by

soil thermometer at the soil depth of 5 cm (whenever applicable). The soil temperature was measured between 10am and 4pm. To account for the inherent hourly and daily temperature variations, we also measured soil temperatures at two fixed places at the hourly basis and used them as the baseline temperature data to correct such sources of uncertainty. The corrected values would be used to compare the difference in mean soil temperature between burned and unburned areas. On the same day the soil samples were collected, 7.5 g fresh organic soil passed through 5 mm sieve and 30 g fresh mineral soil passed through 2 mm sieve were extracted by 75 ml 2 M KCl solution, shaking for 1 h at 160 r/m and then filtered. The extracts were frozen and maintained at -20 ℃ until later laboratory analysis. For each plant species, the foliage was sampled from at least 5 separate individuals within the same plot.

In early June 2015, we further collected plant and soil samples in the same plots. In the unburned area, the dominant overstory species is *Larix gmelinii*, and the dominant understory species include *Vaccinium vitis-idaea*, *Ledum palustre*, *Rhododendron dauricum*, and *Pinus pumila*. In the burned area, the dominant species include seedlings of *Larix gemlinii* and some shrubs and herbs, such as *Vaccinium vitis-idaea*, *Ledum palustre*, *Carex schmidtii* and *Rubus sachalinensis*. Different moss species were observed in unburned and burned area, *Hypnum spp*. was observed in the unburned area, whereas *Polytrichum piliferum* was the common moss species in the burned area. For plant tissues, foliage and branch were sampled from at least 5 separate individuals. Mosses were collected at five random

locations within each plot and were composited. Fine roots were separated from forest floor (Oa+e layer)

and different mineral soil profiles (0-10 cm, 10-20 cm). For soils, forest litter (Oi), forest floor (Oa+e

layer) and three mineral soil profiles (0-10 cm, 10-20 cm and 20-30 cm) samples were collected using

the same method as the one used in year 2014.

**2.3 Laboratory treatment and chemical analysis**

Subsamples of organic and mineral soils were air-dried, crushed and sieved through 5 mm and 2 mm

mesh, respectively, for chemical analysis. Soil pH was measured in $H_2O$ employing a soil: solution ratio

of 1:10. Soil samples were dried at 105 ℃ for 48 h to measure soil water content. Ammonium in the

extract was determined by the indophenol blue method followed by colorimetry, and $NO_3^-$ was

determined colorimely using the same autoanalyser in the form of $NO_2^-$ after reduction of $NO_3^-$ in a

Cd-Cu column followed by the reaction of $NO_2^-$ with N-1-napthylethylenediamine to produce a

chromophore (Rivas *et al.*, 2012).

Plant and soil samples were dried at 60 ℃ to constant weight and ground into powder using a ball

mill and used to analyze $^{15}N$ natural abundance (expressed as $\delta^{15}N$), N and C concentrations by

elemental analyzer (vario MICRO cube; Elementar Analysensysteme GmbH, Hessen Hanau, Germany)

coupled to an IsoPrime100 continuous flow IRMS instrument. Calibrated glycine ($\delta^{15}N$ = 1.6‰),

D-glutamic ($\delta^{15}N$ = -5.7‰), L-histidine ($\delta^{15}N$ = -7.6‰), and acetanilide ($\delta^{15}N$ = 1.4‰) were used as the

internal standards. The $\delta^{15}N$ of the sample relative to the standard (atmospheric $N_2$) was expressed as the following:

$\delta^{15}N = [(R_{sample}/R_{standard}) - 1] * 1000;$

where $R_{sample}$ represents the isotope ratio ($^{15}N/^{14}N$) of sample and $R_{standard}$ is the $^{15}N/^{14}N$ for atmospheric $N_2$. The analytical precision for $\delta^{15}N$ was in general better than 0.2‰.

In order to examine the net N mineralization rate, we collected soil from the same plots in early August of 2014 with the same sampling method. We used the soil samples collected in the late growing season for this purpose because we didn't have low-temperature sample transportation facilities during the first-round soil collection in early June. Net N mineralization rates were estimated using laboratory soil incubations. 7.5 g fresh organic soil passed through 5 mm sieve and 30 g fresh mineral soil passed through 2 mm sieve were put into a plastic cups with polyethylene film to minimize moisture evaporation and incubated at 20 ℃ for 1 week without light. Incubated soil mineral N ($NH_4^+$ and $NO_3^-$) was extracted and measured as above mentioned. On the same day, the soil samples were extracted by 75 ml 2 M KCl solution by the same method as used in the June. The extracts were frozen and maintained at -20 ℃ until later laboratory analysis. Net N mineralization potentials were calculated as the difference between final and initial inorganic N ($NH_4^+ + NO_3^-$) concentrations divided by the

number of incubation days. The expression "N mineralization potential" is used to designate soil samples that produced net amounts of inorganic N.

**2.4 Statistical analysis**

We used one-way analyses of variance (ANOVA) to test whether wildfire significantly affected soil N availability and examine the differences of $\delta^{15}$N (‰) of foliage, organic soil and mineral soil in burned and unburned area. Significance level was set at a *P* value of 0.05 unless otherwise stated. Significant differences among treatment means of soil properties were analyzed using One- way ANOVA. Data were statistically analyzed in R (R Core Team, 2014).

**3 Results**

**3.1 Basic soil properties**

The alteration of soil basic properties in the burned area 4 years after the wildfire was mainly found in the organic soil, not in the mineral soil (Table 1). The soil water content (SWC), total nitrogen (TN), total carbon (TC), C:N were lower in the burned soil, but only the reduction of SWC and TC reached the significant level ($p \leq 0.05$). Mean soil water content at the organic layer was significantly lower in the burned area when compared to the unburned area (41.2% vs. 117.6%). Mean TC at the organic layer in the burned area was 9.2%, which was significantly lower than that in the unburned area (29.2%). In

contrast of those properties that were reduced after fire, pH and temperature were increased. The mean

organic soil temperature in the burned area was 10.0 $^{\circ}$C and was significantly higher than that in the

unburned area (2.9 $^{\circ}$C).

**3.2 Soil inorganic nitrogen concentrations**

At the beginning of growing season (early June), total inorganic N pools were greater in the organic soil

than the mineral soil both in burned and unburned area (Fig. 2A). However, the significant increases in

soil inorganic N concentrations in response to wildfire were only observed in the mineral soil. Mean

total inorganic N concentration in the mineral soil in the burned area (5.55 mg N $kg^{-1}$) was significantly

higher than that (2.22 mg N $kg^{-1}$) in the unburned area. Compared to unburned area (1.6 mg N $kg^{-1}$), the

amount of $NH_4^+$ was significantly higher (5.0 mg N $kg^{-1}$) in mineral soil of burned area (Fig. 2B). $NO_3^-$

concentrations were consistently low in both organic and mineral soil, and had no difference between

burned and unburned area (Fig. 2C).

At the end of growing season (early August), there were no significant differences in total inorganic

N, ammonium and nitrate concentrations between burned and unburned area (Figs. 2D and 2F).

However, the significant decreases in soil inorganic N concentrations in response to wildfire were only

observed in the mineral soil. Mean total inorganic N and ammonium concentrations in the organic soil

were 5.86 and 4.27 mg N $kg^{-1}$ in the burned area, which were significantly lower than those (12.07 and

220    10.35 mg N $kg^{-1}$, respectively) in the unburned area (Figs. 2D and 2E).

**3.3 Nitrogen transformation rates**

The response pattern of N transformation after the fire was similar to that of soil inorganic N concentrations. Both mean net mineralization and ammonification rates (0.056 mg N $kg^{-1}d^{-1}$ and 0.029 mg N $kg^{-1}d^{-1}$) were significantly higher in the organic soil of the burned area compared to the unburned

225    area  of -0.653 mg N $kg^{-1}d^{-1}$ and -0.579 mg N $kg^{-1}d^{-1}$, respectively (Figs. 3A and 3B). In contrast, ANOVA test revealed no significant differences on mineralization and ammonification rates between burned and unburned mineral soil. There was no significant difference in net nitrification either in the organic or in the mineral soil between burned and unburned soil (Fig. 3C).

**3.4 Plant and soil $\delta^{15}N$**

[revised manuscript text omitted]

**4.2 The effect of fire on soil $\delta^{15}N$**

Our results showed that [15]N natural abundance in organic soil was significantly higher in the burned area than unburned area (Fig. 4). These results are consistent with our expectation that soil $\delta^{15}N$ would be higher in the burned area than unburned area, especially in the organic soil. Similar results were reported in other forest ecosystems (LeDuc *et al*., 2013; Schafer and Mack, 2010). Combustion of the upper $\delta^{15}N$-depleted surface soil layer and enhanced nitrification are the two widely-recognized mechanisms to explain [15]N enrichment in organic soil (Hogberg, 1997; LeDuc *et al*., 2013; Schafer and Mack, 2010; Szpak, 2014). However, other mechanism such as $NH_3$ volatilization, combustion, litter return, nitrate leaching, denitrification can also contribute to the observed [15]N enrichment in our study.

Our results showed higher net mineralization and net ammonification didn't lead to higher ammonium concentrations in burned organic soils (Figs. 2-3). On the contrary, the ammonium and total

inorganic nitrogen in the organic soil of the burned area were significantly lower than those in the

305    unburned area (Fig. 2). Such lower $NH_4^+$ and TIN concentrations in the burned soil were likely due to

NH_3 volatilization -- although several other mechanisms such as surface run-off and filtration to mineral

soil might also contribute to this observed pattern. Higher soil temperature and pH values in the burned

area as observed in the present study could enhance $NH_3$ volatilization (Nelson and Conrad, 1982;

Raison, 1979). $NH_3$ volatilization is associated with strong fractionation against $^{15}N$ and higher gaseous

310    losses of $^{15}N$-depleted $NH_3$, and leads to the remaining soil $NH_4^+$ to be enriched in $^{15}N$ (Hobbie and

Ouimette, 2009). Therefore, fire-stimulated $NH_3$ volatilization, associated with strong isotopic

fractionation and subsequent export of $^{15}N$-deplted $NO_3^-$, is considered as one likely being responsible

for $^{15}N$ enrichment of organic soil.

Combustion of surface soil layer could also cause the upper soil to be enriched in $^{15}N$ since high,

315    sustained fire temperatures cause a greater loss of $^{14}N$ compared to $^{15}N$ (Huber $et\ al.$, 2013; Schafer and

Mack, 2010). In our study, wildfire, characterized with high temperature, combusted the thick organic

layer, leading to a significant higher $\delta^{15}N$ in the burned organic soil. For the mineral soil, the significant

higher $\delta^{15}N$ was only observed in 0-10cm mineral soil but not in deeper mineral soil. This pattern may

have resulted from the insulation of underlying mineral soil from heating and limited downward

320    conduction of heat from soil surface to deep soil (Smithwick, $et\ al.$, 2005).

The $^{15}$N enriched litter return, to some extent, might have an effect on $^{15}$N enrichment in the upper soil in the burned area. Plant tissues fallen onto the surface soil, resulting in litter with a similar value of $\delta^{15}$N. In mature larch boreal forest where N is limited, the $^{15}$N-depleted leaf could lead to a lower $\delta^{15}$N in litter. In the burned area, Oi was supposed to have a similar $\delta^{15}$N with the leaf. However, the Oi was composed of a large number of $^{15}$N-depleted coarse woody debris and a small number of recently added litter with a higher $\delta^{15}$N value, which contribute to a relatively lower $\delta^{15}$N in Oi than that in leaf in the burned area.

Fang *et al.* (2015) reported that denitrification was an important N loss pathway and could account for 48% to 86% total $NO_3^-$ loss in forest ecosystems. Although we didn't measure this N process directly, we also considered denitrification as a potential mechanism for the higher $\delta^{15}$N in the organic soil. On one hand, lower plant and microbial biomass in the burned area would result in lower N need and more $NO_3^-$ loss through denitrification. On the other hand, the lack of increase in net nitrification of the burned soil resulted from 7-day laboratory incubation might be due to an enhanced denitrification, which is associated with strong fractionation against $^{15}$N and higher gaseous losses of $^{15}$N -depleted $N_2$ or $N_2O$, remaining soil $NO_3^-$ to be enriched in $^{15}$N (Hobbie and Ouimette, 2009; Robinson, 2001).

**4.3 The effect of fire on plant $\delta^{15}$N**

Foliar $\delta^{15}N$ values were significantly higher in the burned area, which supports our initial expectation that plant $\delta^{15}N$ in the burned forest would be enriched in $^{15}N$. Three complementary processes are likely responsible for this $^{15}N$ enrichment. First, fire consumed the $^{15}N$-depleted surface layers of litter, forcing plants to take up the N from deeper horizons which are more enriched $^{15}N$ than the surface soil (Hogberg, 1997; Sah *et al*., 2006). This assumption is supported by our field experiment in 2015. We found the root was significantly lower in the organic layer and fine roots were mainly distributed in the 0-20 cm mineral soil in the burned area; while in the unburned area, fine roots were mainly distributed in the organic soil layer (Appendix 2). Secondly, part of $^{15}N$-enriched $NH_4^+$ and $NO_3^-$ infiltrated into the deeper mineral soil with rainfall from the organic layer, which leads to the remaining soil N pool to be enriched in $^{15}N$ and further expressed in the values of $\delta^{15}N$ in plant that utilized these N pools for their N demand. Thirdly, increased N availability could lead to a lower dependence of plant N nutrition upon mycorrhizal fungi, which provide their host plants with $^{15}N$-depleted N relative to the soil N sources (Craine *et al*., 2009; Hobbie *et al*., 2008). Boreal forest is a typical N-limited ecosystem and plants usually associated with mycorrhizal fungi to meet their N demand (Craine *et al*., 2009; Hobbie *et al*., 2008; Nasholm *et al*., 2013). Larch is the dominant species in the unburned area and is often associated with ECM. *Ledum* spp. and *Vaccinium* spp. are the main understory species and are often associated with ERM (Michelsen *et al*., 1998). Numerous studies have suggested that mycorrhizal fungi

340

345

350

355   preferentially transfer isotopically depleted nitrogen to their host plants (Hobbie and Agerer, 2010;

Högberg 1997; Whiteside *et al*., 2012). Thus we considered the mycorrhizal fungi would play a key role

in N supply for plant in the unburned area and lead to lower foliar $\delta^{15}N$ values of their host plants.

*Vaccinium vitis-idaea, Ledum palustre* and *Deyeuxia angustifolia* were species occurring in both

burned and unburned area. Nevertheless, there were significant differences in their foliar $\delta^{15}N$ values

360   between burned and unburned area. For example, *Vaccinium vitis-idaea* $\delta^{15}N$ values were 0.2‰ and

-3.7‰, respectively, in burned and unburned area (Table 2). Different N resources and change of fine

roots distribution induced by fire could contribute these differences. Moreover, compared to the

unburned area (mature larch boreal forest), which is a typical N limited ecosystem and has a negative

foliar $\delta^{15}N$ (-3.7‰), the plant has a higher $\delta^{15}N$ in burned area, suggesting that this ecosystem has

365   shifted from N limited to N open.

**5 Conclusions**

In this study we demonstrated that wildfire had a profound influence on N cycles in the boreal forests of

the Great Xing'an Mountains. The ecosystem N cycle was still open in 4 and 5 years after fire. However,

the wildfire effects were mainly limited in organic layer and 0-10cm mineral soil. The fire-induced

370   increases in net mineralization rate and net ammonification rate were only exhibited in the organic soil,

not in the mineral soil. The increased organic layer temperature and pH, decreased moisture and C:N

could be the primary mechanism determining inorganic N transformation rates. We suggest that the observed $^{15}$N enrichment in soil might be attributed to various mechanisms such as $NH_3$ volatilization, combustion, litter return and denitrification. Greater dependence of plant on deeper soil N and less dependence on mycorrhizal fungi might increase the $^{15}$N of plant in the burned area. The $\delta^{15}$N of plant and soil could be considered as a comprehensive indicator for explore the responses of N processes to wildfire in forest ecosystems.

**Acknowledgement**

This research was supported by the National Natural Science Foundation of China (31270511, 41222004, 31422009, 41301200) and State Key Laboratory of Forest and Soil Ecology, Institute of Applied Ecology, the Chinese Academy of Sciences (No. LFSE 2013-13). We acknowledge Jiaxing Zu and Yue Yu for their assistance in the field work. We thank staff in Huzhong National Natural Reserve for their supports in the field sampling. We also appreciate Prof. Gundersen's helpful suggestion on revising the manuscript.

Table 1. Basic soil properties at two soil layer in unburned (n=12) and burned area (n=24). Values presented are means with the standard error in parentheses. Means in a row that have the same letter are not significantly different at alpha level is 0.05 (ANOVA, $p \leqslant 0.05$).

| Layer | SWC (%) | | pH | | TN (%) | | TC (%) | | C:N | | T(℃) | |
|---|---|---|---|---|---|---|---|---|---|---|---|---|
| | Unburned | Burned | Unburned | Burned | Unburned | Burned | Unburned | Burned | Unburned | Burned | Unburned | Burned |
| Organic layer | 117.6 (32.0)a | 41.2 (18.6)b | 4.4 (0.5)a | 5.2 (0.5)a | 0.8 (0.5)a | 0.4 (0.2)a | 29.2 (10.2)a | 9.2 (4.0)b | 27.5 (5.0)a | 20.8 (5.4)a | 2.9 (1.1)a | 10.0 (2.5)b |
| Mineral layer (0-20cm) | 36.2 (8.0)a | 35.2 (7.7)a | 5.2 (0.4)a | 5.3 (0.3)a | 0.2 (0.0)a | 0.2 (0.0)a | 4.3 (1.1)a | 3.1 (0.5)a | 24.1 (4.1)a | 20.3 (2.4)a | NA | NA |

Table 2. Foliar stable N isotope ratio ($\delta^{15}$N), N concentration, C concentration and C:N ratios for each sampled species in burned and unburned area.

| Site location | Species | $\delta^{15}$N (‰) | N conc.(%) | C conc.(%) | C:N ratio |
|---|---|---|---|---|---|
| Burned area | *Vaccinium vitis-idaea* | 0.2 | 1.4 | 50.1 | 36.1 |
| | *Ledum palustre* | 2.6 | 2.1 | 51.8 | 25.2 |
| | *Deyeuxia angustifolia* | 1.8 | 2.6 | 43.7 | 17.1 |
| | *Carex schmidtii* | 3.4 | 2.1 | 42.9 | 21.4 |
| | *Chamerion angustifolium* | 5.2 | 3.9 | 46 | 12 |
| | *Betula platyphylla* | 2.4 | 3.1 | 48 | 15.7 |
| | *Rubus sachalinensis* | 3.4 | 2.8 | 44.5 | 16 |
| | Mean±SE | 3.7±1.9 | 2.9±0.9 | 45.5±2.6 | 17.3±5.5 |
| Unburned area | *Vaccinium vitis-idaea* | -3.7 | 1.2 | 49.4 | 40.9 |
| | *Ledum palustre* | -3.4 | 2 | 51.5 | 26.3 |
| | *Deyeuxia angustifolia* | -2.4 | 2.3 | 42.7 | 18.3 |
| | *Pinus pumila* | -4.2 | 1.3 | 49.3 | 39.8 |
| | *Larix gmelini* | -4.6 | 1.6 | 48.2 | 31.2 |
| | *Rhododendron dauricum* | -2.9 | 2.1 | 48 | 22.6 |
| | Mean±SE | -3.7±1.3 | 1.7±0.5 | 48.7±2.0 | 31.5±9.0 |

[Figure]

Figure 1. Map of research (burned and unburned) sites in the Huzhong Natural Reserve (HNR), China. A large wildfire burned almost 600 ha mature larch forest within the HNR in the summer of 2010. The red boundary represents the burned area. Unburned area is chosen in nearby burned area as a control. The black triangles represent burned plots, the yellow circles represent unburned plots.

[Figure]

Figure 2. Soil inorganic N concentrations (TIN, $NO_3^-$-N and $NH_4^+$-N)) of organic and 0-20cm mineral soils sampled in June, 2014 (A, B,C) and in August, 2014 (D, E,F). Bars show means $\pm$ standard error. Different letters indicate significant difference between burned and unburned plots.

[Figure]

Figure 3. N transformation rates (net mineralization, ammonification and nitrification) of organic and mineral soils (0-20 cm) sampled in August, 2014.

Bars show means ± standard error. Different letters indicate significant difference between burned and unburned plots.

[Figure]

Figure 4. $\delta^{15}$N (‰) for plant and soil in unburned and burned systems 5 years after wildfire. Solid red circles represent burned plots, solid black squares represent unburned plots, respectively. Data show means ± standard error. One asterisk indicate significant difference among forests at $p \leq 0.05$, two asterisks indicate significant difference among forests at $p \leq 0.01$, three asterisks indicate significant difference among forests at $p \leq 0.001$.

Appendix 1. $\delta^{15}$N values (‰) of foliage, organic soil and mineral soils 4 years after wildfire. Values presented are means with the standard error in parentheses. Means in a row that have the same letter are not significantly different at alpha level is 0.05 (ANOVA, $p \leqslant 0.05$).

| N pool | $\delta^{15}$N (‰) | |
|---|---|---|
| | Unburned | Burned |
| Foliar | -3.7(1.3)b | 3.7(1.9)a |
| Organic soil | 1.3(1.2)b | 3.6(0.8)a |
| Mineral soil(0-20cm) | 4.8(0.3)a | 4.9(0.6)a |

Appendix 2. The fine roots biomass in two soil layers of both burned and unburned area. Means in a row that have the same letter are not significantly different at alpha level is 0.05 (ANOVA, $p \leqslant 0.05$).

| Layer | Fine root (kg ha$^{-1}$) | |
|---|---|---|
| | Unburned | Burned |
| Organic layer | 24 405(13 503)a | 1 054(1 824)b |
| Mineral layer(0-20cm) | 4 826(9 037)a | 6 275(1 595)a |

---

## Author Comment (AC2) · 27 Jun 2016

We appreciate the reviewer 2's professional comments which helped us to improve our manuscript. According to the comments we modified the manuscript as detailed below.

Reviewer 2's comments: The "mechanisms" that explain these observations are derived from patterns within the observations, but there are two problems associated with this extrapolation: 1) I was concerned by the fact that the mineralization results are compared to the TIN results, despite the fact that they were taken in different seasons. Specifically, the TIN samples were collected before (June) the wettest part of the year (June to August; Line 107), whereas the N mineralization samples were collected after the wet season (Autumn, but not specified). Though it is well known that the size and

direction of N pools and fluxes can change seasonally, these data are compared to each other as though they represent the same N. For example, the authors posit that the "high soil NH4+ pools did not lead to an elevated net nitrification rate" (Line 274), but the NH4+ pool and the net nitrification rate were collected months apart and likely had little bearing on each other. These two time points should not be compared this way.

→Authors' response: First of all, there was a typo in our previous version. The N mineralization samples were collected in early August, not early Autumn. Still, we accepted referee #2's comment regarding the improper comparison of TIN and N mineralization values that were measured at two different time points. In fact, a similar concern was also raised by reviewer 1. In this revision, we provided new data about inorganic nitrogen concentrations in August to ensure the comparison of these two variables were derived from the samples collected at the same time (Figs 2D-F). We showed the ammonification rate of the organic soil in the burned area was higher than that in the unburned area. To the contrary, the ammonium concentration was significantly lower than that in the unburned area.

Reviewer 2's comments: "2) As a result of the previous comment, the "mechanisms" that could explain these patterns of N cycling are difficult to establish, at best. The authors state that "a large amount of NH4+ was lost through volatilization" (Lines 276-277). However, there is no empirical evidence for this, and no way in which to rule out other mechanisms that explain differences in 15N isotopic signatures between the sites, such as denitrification or combustion (the authors addressed these, but emphasized volatilization as the proximate mechanism controlling 15N)."

→Authors' response: Reviewer 1 also raised this issue. Please see our response to Reviewer 1's general comment.

Reviewer 2's comments: "Throughout, from the title through the discussion, there is a strong emphasis placed on the revelation that $\delta$15N represented an "open" N cycle.

For example, in the introduction, it is stated that "$\delta$15N could provide us a promising and comprehensive tool to detect the effect of wildfire on N cycling" (Lines 57-58). This makes it sound like establishing this is an objective of the paper, but this is a well-established use of these measurements (see Martinelli et al. 1999). By contrast, the important result related to the stated objectives would seem to be that fire leaves an open N cycle for several years. Likely, this problem could be addressed by changing verbiage."

→Authors' response: we have accepted referee #2's comments and changed the wording to "In fact, although the responses of available N concentration and N mineralization to wildfire could vary by time and space, we can expect higher values of $\delta$15N in plant and soil as a legacy of longer or short term opening in the N cycles when available N supply exceeds demand, resulting in an increase in N loss. Therefore, $\delta$15N could provide us a promising and comprehensive tool to detect the legacy effect of wildfire on N cycling openness years after the fire disturbance has occurred."

Reviewer 2's comments: "There was a paucity of information about the fire: it was described as severe (line 118), but little other information was provided than unpublished results, not explained in the methods, that there was a tenfold loss of aboveground biomass (Lines 275-276). I wonder why the authors would expect to see a recovery of the N cycle given such substantial fire-related effects persist? This was not clearly articulated in the manuscript- what was the underlying rationale for this work, other than the lack of data on N cycling from Chinese larch forest?"

→Authors' response: We added more information about the fire as the following. The historical fire regime in this region is described as frequent, surface fires mixed with infrequent, stand-replacing crown fires, with fire-free interval ranged from 30 to 120 years (Xu et al., 1997; Liu et al., 2012). However, climate change, forest management and human activities have altered fire regimes in this region (Jackson et al., 1997; Wang et al., 2007). Although the dominant tree species Dahurian larch is regarded as a fire-tolerant species with thick bark near the stem bottom, its post-fire mortality

rate is still high, mainly due to a horizontal shallow-distributed root system (Fang et al., 2015; Vijayakumar et al., 2016). A stand-replacing wildfireïijŇwhich was ignited by lighting, burned 600 ha of Huzhong National Natural Reserve on June 26th, 2010. This fire provided an ideal opportunity to study the effects of fire on soil N dynamics in this ecosystem." As to the post-fire recovery patterns of the N cycle, extensive meta-analysis and literature review studies have reached a consensus that N availability immediately after fire would be elevated, followed by a rapid decline within the first few years (Wan et al., 2001; Smithwick et al., 2005; Wang et al., 2014). Such short-term N cycling responses to wildfire have been widely documented in grassland and temperate forest ecosystems that are dominated by low-severity wildfires. For the forest ecosystems dominated by high-severity fires, the effects of wildfire on N cycling remain controversial. Some studies reported N availability would return to unburned level after several years since the wildfires (Turner et al., 2007; Koyama et al., 2010), while others documented the persistence of elevated N availability for several decades (Deluca et al., 2002; Kurth et al., 2014). Notably, all of these studies occurred in US and North European forests that experienced stand-replacing fires. In contrast, there is a scarcity of such studies in Eurasian boreal forest. Our research group has previously investigated two fires burned in the Chinese boreal forest and found that N availability in one-year-after-burn stands was significantly higher than unburned stands, but such elevated N availability was not detected in 11-yr-after-burn stands (Kong et al., 2015). In light of literature and our previous work, we hypothesize that the inorganic nitrogen concentrations and N mineralization rates in the burned area would be similar with the unburned area.

Reviewer 2's comments: "I was also surprised that there was no mention of the role of cation exchange capacity in N cycling. While fluxes of N were affected by fire in the litter, N pools were affected in the soil. This likely represents the simple fact that a changed physical environment, combined with a reduction in plant uptake, means greater microbial processing of N in the organic layer and greater sorption of N in the mineral soil. This wasn't clearly articulated despite the fact that several lines were

dedicated to changes in temperature (Lines 239-244)."

→Authors' response: Thanks for the excellent suggestion. In this revision, we discussed how changes of biotic and environmental conditions could contribute to the increase of N mineralization. We pointed out that the increased cation exchange capacity and pH resulted from ash deposition, and the increased temperature resulted from decreased thickness of organic layer would promote microbe activities, which could result in higher mineralization in the field. The resulting increased N mineralization in the organic soil, coupled with reduction of plant uptake, could further lead to greater sorption of N in the mineral soil. We made the following changes: Post-fire abiotic environments such as increased soil pH resulted from increased base cation availability and temperature (Table 1) tend to be more suitable for microbial activities (Smithwick et al., 2005). Increased temperature might have played a key role in N transformation (Klopatek et al.,1990) because decomposition rates may increase by 50% - 100% when soil temperatures increase 5 oC – 10 oC (Richter et al., 2000).

Reviewer 2's comments: Some specific comments: There were a number of typos and awkward phrases that would need to be cleared up before publication (for example: "winder" instead of winter (Line 106), "sever" instead of severe (Line 118), "colorimely" instead of colorimetrically (Line 146), "filtrated" instead of filtered (Line 131), "burned polts" instead of burned plots (Table 2) – not a complete list).

→Authors' response: We appreciate that the referee #2 pointed out the problems in English presentation and have thoroughly checked our spelling.

Reviewer 2's comments: There are better citations for studies suggesting N limits high latitude terrestrial ecosystems than Popova et al. 2013 and Stark & Hart 1997 (Lines 34-35). Consider Vitousek & Howarth 1991, Lebauer and Treseder 2008, Elser et al. 2007, and Harpole et al. 2011).

→Authors' response: we have accepted referee #2's suggestion and used the citations of "(Elser et al. 2007, Harpole et al. 2011; Lebauer and Treseder, 2008; Vitousek and

Howarth 1991)" to replace the citations of "(Popova et al., 2013; Stark and Hart, 1997)".

Reviewer 2's comments: How is Figure 3 different from Figure 4? Similarly, how are the values provided in the paragraph beginning on Line 202 different from the values described in the paragraph beginning on Line 211? Both involve foliar $\delta$15N, for example. Not clear.

→Authors' response: We agree these two figures seemed redundant. Therefore, we kept Figure 4 with more detailed information of N pool for different plant tissue and soil profiles, and we presented the data of Figure 3 as the Appendix 1.

Reviewer 2's comments: "How was the branch, moss and fine root 15N relevant to the objectives? Seems like these data, especially the moss, are ancillary and it was not clearly stated that there was a goal to observe vertical profiles of $\delta$15N signatures from tree top to 30 cm mineral soil."

→Authors' response: Figure 4 showed the vertical profiles of $\delta$15N signatures from tree top to 30 cm mineral soil. We found that wildfire had a more significant effect on plant than soil and the effect of wildfire on soil would decrease with the soil depth. The Figure 4 also showed the litter return might be a potential mechanism for the 15N enrichment in organic soil.

Reviewer 2's comments: "Figure 1 didn't come out very well, and wasn't very helpful. The caption makes it sound like the unburned area is in the burned area: "Unburned area is chosen in the nearby burned area as control." →Authors' response: We appreciate that the referee #2 kindly pointed out the problems in Figure caption. We changed the caption to "Research locations in the Huzhong Natural Reserve (HNR), China. A large wildfire burned almost 600 ha mature larch forest within the HNR in the summer of 2010. The red boundary represents the burned area. The black triangles represent burned plots, the yellow circles represent unburned plots which were established in the unburned area as control." We also added the sketch of the Huzhong Natural Reserve (HNR) and the locations of 2010 burned area in Figure 1.